# Testing Unfaithful Gaussian Graphical Models

**De Wen Soh**
Department of Electrical Engineering
Yale University
17 Hillhouse Ave, New Haven, CT 06511
dewen.soh@yale.edu

**Sekhar Tatikonda**
Department of Electrical Engineering
Yale University
17 Hillhouse Ave, New Haven, CT 06511
sekhar.tatikonda@yale.edu

## Abstract

The global Markov property for Gaussian graphical models ensures graph separation implies conditional independence. Specifically if a node set $S$ graph separates nodes $u$ and $v$ then $X_u$ is conditionally independent of $X_v$ given $X_S$. The opposite direction need not be true, that is, $X_u \perp X_v \mid X_S$ need not imply $S$ is a node separator of $u$ and $v$. When it does, the relation $X_u \perp X_v \mid X_S$ is called faithful. In this paper we provide a characterization of faithful relations and then provide an algorithm to test faithfulness based only on knowledge of other conditional relations of the form $X_i \perp X_j \mid X_S$.

## 1 Introduction

Graphical models [1, 2, 3] are a popular and important means of representing certain conditional independence relations between random variables. In a Gaussian graphical model, each variable is associated with a node in a graph, and any two nodes are connected by an undirected edge if and only if their two corresponding variables are independent conditioned on the rest of the variables. An edge between two nodes therefore corresponds directly to the non-zero entries of the precision matrix $\mathbf{\Omega} = \mathbf{\Sigma}^{-1}$, where $\mathbf{\Sigma}$ is the covariance matrix of the multivariate Gaussian distribution in question. With the graphical model defined in this way, the Gaussian distribution satisfies the global Markov property: for any pair of nodes $i$ and $j$, if all paths between the two pass through a set of nodes $S$, then the variables associated with $i$ and $j$ are conditionally independent given the variables associated with $S$.

The converse of the global Markov property does not always hold. When it does hold for a conditional independence relation, that relation is called faithful. If it holds for all relations in a model, that model is faithful. Faithfulness is important in structural estimation of graphical models, that is, identifying the zeros of $\mathbf{\Omega}$. It can be challenging to simply invert $\mathbf{\Sigma}$. With faithfulness, to determine an edge between nodes $i$ and $j$, one could run through all possible separator sets $S$ and test for conditional independence. If $S$ is small, the computation becomes more accurate. In the work of [4, 5, 6, 7], different assumptions are used to bound $S$ to this end.

The main problem of faithfulness in graphical models is one of identifiability. Can we distinguish between a faithful graphical model and an unfaithful one? The idea of faithfulness was first explored for conditional independence relations that were satisfied in a family of graphs, using the notion of $\theta$-Markov perfectness [8, 9]. For Gaussian graphical models with a tree topology the the distribution has been shown to be faithful [10, 11]. In directed graphical models, the class of unfaithful distributions has been studied in [12, 13]. In [14, 15], a notion of strong-faithfulness as a means of relaxing the conditions of faithfulness is defined.

In this paper, we study the identifiability of a conditional independence relation. In [6], the authors restrict their study of Gaussians to walk-summable ones. In [7], the authors restrict their class of distributions to loosely connected Markov random fields. These restrictions are such that the

local conditional independence relations imply something about the global structure of the graphical model. In our discussion, we assume no such restrictions. We provide a testable condition for the faithfulness of a conditional independence relation in a Gaussian undirected graphical model. Checking this condition requires only using other conditional independence relations in the graph. We can think of these conditional independence relations as local patches of the covariance matrix $\mathbf{\Sigma}$. To check if a local patch reflects the global graph (that is, a local path is faithful) we have to make use of other local patches. Our algorithm is the first algorithm, to the best of our knowledge, that is able to distinguish between faithful and unfaithful conditional independence relations without any restrictions on the topology or assumptions on spatial mixing of the Gaussian graphical model.

This paper is structured as follows: In Section 2, we discuss some preliminaries. In Section 3, we state our main theorem and proofs, as well as key lemmas used in the proofs. In Section 4, we lay out an algorithm that detects unfaithful conditional independence relations in Gaussian graphical models using only local patches of the covariance matrix. We also describe a graph learning algorithm for unfaithful graphical models. In Section 5, we discuss possible future directions of research.

## 2   Preliminaries

We first define some linear algebra and graph notation. For a matrix $\mathbf{M}$, let $\mathbf{M}^T$ denote its transpose and let $|\mathbf{M}|$ denote its determinant. If $I$ is a subset of its row indices and $J$ a subset of its column indices, then we define the submatrix $\mathbf{M}_{IJ}$ as the $|I| \times |J|$ matrix with elements with both row and column indices from $I$ and $J$ respectively. If $I = J$, we use the notation $\mathbf{M}_I$ for convenience. Let $\mathbf{M}(-i, -j)$ be the submatrix of $\mathbf{M}$ with the $i$-th row and $j$-th column deleted. Let $\mathbf{M}(-I, -J)$ be the submatrix with rows with indices from $I$ and columns with indices from $J$ removed. In the same way, for a vector $\mathbf{v}$, we define $\mathbf{v}_I$ to be the subvector of $\mathbf{v}$ with indices from $I$. Similarly, we define $\mathbf{v}(-I)$ to be the subvector of $\mathbf{v}$ with indices not from $I$. For two vectors $\mathbf{v}$ and $\mathbf{w}$, we denote the usual dot product by $\mathbf{v} \cdot \mathbf{w}$.

Let $\mathcal{G} = (\mathcal{W}, \mathcal{E})$ be an undirected graph, where $\mathcal{W} = \{1, \ldots, n\}$ is the set of nodes and $\mathcal{E}$ is the set of edges, namely, a subset of the set of all unordered pairs $\{(u, v) \mid u, v \in \mathcal{W}\}$. In our paper we are dealing with graphs that have no self-loops and no multiple edges between the same pair of nodes. For $I \subseteq \mathcal{W}$, we denote the induced subgraph on nodes $I$ by $\mathcal{G}_I$. For any two distinct nodes $u$ and $v$, we say that the node set $S \subseteq \mathcal{W} \setminus \{u, v\}$ is a node separator of $u$ and $v$ if all paths from $u$ to $v$ must pass through some node in $S$.

Let $\mathbf{X} = (X_1, \ldots, X_n)$ be a multivariate Gaussian distribution with mean $\boldsymbol{\mu}$ and covariance matrix $\mathbf{\Sigma}$. Let $\mathbf{\Omega} = \mathbf{\Sigma}^{-1}$ be the precision or concentration matrix of the graph. For any set $S \subset \mathcal{W}$, we define $\mathbf{X}_S = \{X_i \mid i \in S\}$. We note here that $\mathbf{\Sigma}_{uv} = 0$ if and only if $X_u$ is independent of $X_v$, which we denote by $X_u \perp X_v$. If $X_u$ is independent of $X_v$ conditioned on some random variable $Z$, we denote this independence relation by $X_u \perp X_v \mid Z$. Note that $\mathbf{\Omega}_{uv} = 0$ if and only if $X_u \perp X_v \mid \mathbf{X}_{\mathcal{W} \setminus \{u, v\}}$.

For any set $S \subseteq \mathcal{W}$, the conditional distribution of $\mathbf{X}_{\mathcal{W} \setminus S}$ given $\mathbf{X}_S = \mathbf{x}_S$ follows a multivariate Gaussian distribution with conditional mean $\boldsymbol{\mu}_{\mathcal{W} \setminus S} - \mathbf{\Sigma}_{(\mathcal{W} \setminus S)S} \mathbf{\Sigma}_S^{-1} (\mathbf{x}_S - \boldsymbol{\mu}_S)$ and conditional covariance matrix $\mathbf{\Sigma}_{\mathcal{W} \setminus S} - \mathbf{\Sigma}_{(\mathcal{W} \setminus S)S} \mathbf{\Sigma}_S^{-1} \mathbf{\Sigma}_{S(\mathcal{W} \setminus S)}$. For distinct nodes $u, v \in \mathcal{W}$ and any set $S \subseteq \mathcal{W} \setminus \{u, v\}$, the following property easily follows.

**Proposition 1** $X_u \perp X_v \mid \mathbf{X}_S$ if and only if $\mathbf{\Sigma}_{uv} = \mathbf{\Sigma}_{uS} \mathbf{\Sigma}_S^{-1} \mathbf{\Sigma}_{Sv}$.

The concentration graph $\mathcal{G}_{\mathbf{\Sigma}} = (\mathcal{W}, \mathcal{E})$ of a multivariate Gaussian distribution $\mathbf{X}$ is defined as follows: We have node set $\mathcal{W} = \{1, \ldots, n\}$, with random variable $X_u$ associated with node $u$, and edge set $\mathcal{E}$ where unordered pair $(u, v)$ is in $\mathcal{E}$ if and only if $\mathbf{\Omega}_{uv} \neq 0$. The multivariate Gaussian distribution, along with its concentration graph, is also known as a Gaussian graphical model. Any Gaussian graphical model satisfies the global Markov property, that is, if $S$ is a node separator of nodes $u$ and $v$ in $\mathcal{G}_{\mathbf{\Sigma}}$, then $X_u \perp X_v \mid \mathbf{X}_S$. The converse is not necessarily true, and therefore, this motivates us to define faithfulness in a graphical model.

**Definition 1** *The conditional independence relation $X_u \perp X_v \mid \mathbf{X}_S$ is said to be faithful if $S$ is a node separator of $u$ and $v$ in the concentration graph $\mathcal{G}_{\mathbf{\Sigma}}$. Otherwise, it is unfaithful. A multivari-*

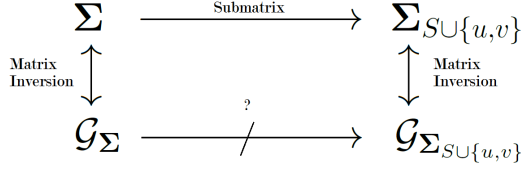

Figure 1: Even though $\mathbf{\Sigma}_{S\cup\{u,v\}}$ is a submatrix of $\mathbf{\Sigma}$, $\mathcal{G}_{\mathbf{\Sigma}_{S\cup\{u,v\}}}$ need not be a subgraph of $\mathcal{G}_{\mathbf{\Sigma}}$. Edge properties do not translate as well. That means the local patch $\mathbf{\Sigma}_{S\cup\{u,v\}}$ need not reflect the edge properties of the global graph structure of $\mathbf{\Sigma}$.

*ate Gaussian distribution is faithful if all its conditional independence relations are faithful. The distribution is unfaithful if it is not faithful.*

**Example 1 (Example of an unfaithful Gaussian distribution)** *Consider the multivariate Gaussian distribution $\mathbf{X} = (X_1, X_2, X_3, X_4)$ with zero mean and positive definite covariance matrix*

$$\mathbf{\Sigma} = \begin{bmatrix} 3 & 2 & 1 & 2 \\ 2 & 4 & 2 & 1 \\ 1 & 2 & 7 & 1 \\ 2 & 1 & 1 & 6 \end{bmatrix}. \tag{1}$$

*By Proposition 1, we have $X_1 \perp X_3 \mid X_2$ since $\mathbf{\Sigma}_{13} = \mathbf{\Sigma}_{12}\mathbf{\Sigma}_{22}^{-1}\mathbf{\Sigma}_{23}$. However, the precision matrix $\mathbf{\Omega} = \mathbf{\Sigma}^{-1}$ has no zero entries, so the concentration graph is a complete graph. This means that node 2 is not a node separator of nodes 1 and 3. The independence relation $X_1 \perp X_3 \mid X_2$ is thus not faithful and the distribution $\mathbf{X}$ is not faithful as well.*

We can think of the submatrix $\mathbf{\Sigma}_{S\cup\{u,v\}}$ as a local patch of the covariance matrix $\mathbf{\Sigma}$. When $X_u \perp X_v \mid \mathbf{X}_S$, nodes $u$ and $v$ are not connected by an edge in the concentration graph of the local patch $\mathbf{\Sigma}_{S\cup\{u,v\}}$, that is, we have $(\mathbf{\Sigma}_{S\cup\{u,v\}}^{-1})_{uv} = 0$. This does not imply that $u$ and $v$ are not connected in the concentration graph $\mathcal{G}_{\mathbf{\Sigma}}$. If $X_u \perp X_v \mid \mathbf{X}_S$ is faithful, then the implication follows. If $X_u \perp X_v \mid \mathbf{X}_S$ is unfaithful, then $u$ and $v$ may be connected in $\mathcal{G}_{\mathbf{\Sigma}}$ (See Figure 1).

Faithfulness is important in structural estimation, especially in high-dimensional settings. If we assume faithfulness, then finding a node set $S$ such that $X_u \perp X_v \mid \mathbf{X}_S$ would imply that there is no edge between $u$ and $v$ in the concentration graph. When we have access only to the sample covariance instead of the population covariance matrix, if the size of $S$ is small compared to $n$, the error of computing $X_u \perp X_v \mid \mathbf{X}_S$ is much less than the error of inverting the entire covariance matrix. This method of searching through all possible node separator sets of a certain size is employed in [6, 7]. As mention before, these authors impose other restrictions on their models to overcome the problem of unfaithfulness. We do not place any restriction on the Gaussian models. However, we do not provide probabilistic bounds when dealing with samples, which they do.

## 3 Main Result

In this section, we will state our main theoretical result. This result is the backbone for our algorithm that differentiates a faithful conditional independence relation from an unfaithful one. Our main goal is to decide if a conditional independence relation $X_u \perp X_v \mid \mathbf{X}_S$ is faithful or not. For convenience, we will denote $\mathcal{G}_{\mathbf{\Sigma}}$ simply by $\mathcal{G} = (\mathcal{W}, \mathcal{E})$ for the rest of this paper. Now let us suppose that it is faithful; $S$ is a node separator for $u$ and $v$ in $\mathcal{G}$. Then we should not be able to find a path from $u$ to $v$ in the induced subgraph $\mathcal{G}_{\mathcal{W}\setminus S}$. The main idea therefore is to search for a path between $u$ and $v$ in $\mathcal{G}_{\mathcal{W}\setminus S}$. If this fails, then we know that the conditional independence relation is faithful.

By the global Markov property, for any two distinct nodes $i, j \in \mathcal{W} \setminus S$, if $X_i \not\perp X_j \mid \mathbf{X}_S$, then we know that there is a path between $i$ and $j$ in $\mathcal{G}_{\mathcal{W}\setminus S}$. Thus, if we find some $w \in \mathcal{W} \setminus (S \cup \{i, j\})$ such that $X_u \not\perp X_w \mid \mathbf{X}_S$ and $X_v \not\perp X_w \mid \mathbf{X}_S$, then a path exists from $u$ to $w$ and another exists from $v$ to $w$, so $u$ and $v$ are connected in $\mathcal{G}_{\mathcal{W}\setminus S}$. This would imply that $X_u \perp X_v \mid \mathbf{X}_S$ is unfaithful.

However, testing for paths this way does not necessarily rule out all possible paths in $\mathcal{G}_{\mathcal{W}\setminus S}$. The problem is that some paths may be obscured by other unfaithful conditional independence relations. There may be some $w$ whereby $X_u \not\perp X_w \mid \boldsymbol{X}_S$ and $X_v \perp X_w \mid \boldsymbol{X}_S$, but the latter relation is unfaithful. This path from $u$ to $v$ through $w$ is thus not detected by these two independence relations.

We will show however, that if there is no path from $u$ to $v$ in $\mathcal{G}_{\mathcal{W}\setminus S}$, then we cannot find a series of distinct nodes $w_1, \ldots, w_t \in \mathcal{W} \setminus (S \cup \{u,v\})$ for some natural number $t > 0$ such that $X_u \not\perp X_{w_1} \mid \boldsymbol{X}_S, X_{w_1} \not\perp X_{w_2} \mid \boldsymbol{X}_S, \ldots, X_{w_{t-1}} \not\perp X_{w_t} \mid \boldsymbol{X}_S, X_{w_k} \not\perp X_v \mid \boldsymbol{X}_S$. This is to be expected because of the global Markov property. What is more surprising about our result is that the converse is true. If we cannot find such nodes $w_1, \ldots, w_t$, then $u$ and $v$ are not connected by a path in $\mathcal{G}_{\mathcal{W}\setminus S}$. This means that if there is a path from $u$ to $v$ in $\mathcal{G}_{\mathcal{W}\setminus S}$, even though it may be hidden by some unfaithful conditional independence relations, ultimately there are enough conditional dependence relations to reveal that $u$ and $v$ are connected by a path in $\mathcal{G}_{\mathcal{W}\setminus S}$. This gives us an equivalent condition for faithfulness that is in terms of conditional independence relations.

Not being able to find a series of nodes $w_1, \ldots, w_t$ that form a string of conditional dependencies from $u$ to $v$ as described in the previous paragraph is equivalent to the following: we can find a partition $(U, V)$ of $\mathcal{W} \setminus S$ with $u \in U$ and $v \in V$ such that for all $i \in U$ and $j \in V$, we have $X_i \perp X_j \mid \boldsymbol{X}_S$. Our main result uses the existence of this partition as a test for faithfulness.

**Theorem 1** *Let $\boldsymbol{X} = (X_1, \ldots, X_n)$ be a Gaussian distribution with mean zero, covariance matrix $\boldsymbol{\Sigma}$ and concentration matrix $\boldsymbol{\Omega}$. Let $u, v$ be two distinct elements of $\mathcal{W}$ and $S \subset \mathcal{W} \setminus \{i,j\}$ such that $X_u \perp X_v \mid \boldsymbol{X}_S$. Then $X_u \perp X_v \mid \boldsymbol{X}_S$ is faithful if and only if there exists a partition of $\mathcal{W} \setminus S$ into two disjoint sets $U$ and $V$ such that $u \in U$, $v \in V$, and $X_i \perp X_j \mid \boldsymbol{X}_S$ for any $i \in U$ and $j \in V$.*

*Proof of Theorem 1* . One direction is easy. Suppose $X_u \perp X_v \mid \boldsymbol{X}_S$ is faithful and $S$ separates $u$ and $v$ in $\mathcal{G}$. Let $U$ be the set of all nodes reachable from $u$ in $\mathcal{G}_{\mathcal{W}\setminus S}$ including $u$. Let $V = \{\mathcal{W} \setminus S \cup U\}$. Then $v \in V$ since $S$ separates $u$ and $v$ in $\mathcal{G}$. Also, for any $i \in U$ and $j \in V$, $S$ separates $i$ and $j$ in $\mathcal{G}$, and by the global Markov property, $X_i \perp X_j \mid \boldsymbol{X}_S$.

Next, we prove the opposite direction. Suppose that there exists a partition of $\mathcal{W} \setminus S$ into two sets $U$ and $V$ such that $u \in U$, $v \in V$, and $X_i \perp X_j \mid \boldsymbol{X}_S$. for any $i \in U$ and $j \in V$. Our goal is to show that $S$ separates $u$ and $v$ in the concentration graph $\mathcal{G}$ of $\boldsymbol{X}$. Let $\boldsymbol{\Omega}_{\mathcal{W}\setminus S} = \boldsymbol{\Omega}'$ where the latter is the submatrix of the precision matrix $\boldsymbol{\Omega}$. Let the $h$-th column vector of $\boldsymbol{\Omega}'$ be $\boldsymbol{\omega}^{(h)}$, for $h = 1, \ldots, |\mathcal{W} \setminus S|$.

*Step 1:* We first solve the trivial case where $|U| = |V| = 1$. If $|U| = |V| = 1$, then $S = \mathcal{W} \setminus \{u, v\}$, and trivially, $X_u \perp X_v \mid \boldsymbol{X}_{\mathcal{W}\setminus\{u,v\}}$ implies $S$ separates $u$ and $v$, and we are done. Thus, we assume for the rest of the proof that $U$ and $V$ cannot both be size one.

*Step 2:* We deal with a second trivial case in our proof, which is the case where $\boldsymbol{\omega}^{(i)}(-i)$ is identically zero for any $i \in U$. In the case where $i = u$, we have $\boldsymbol{\Omega}_{uj} = 0$ for all $j \in \mathcal{W} \setminus (S \cup \{u\})$. This implies that $u$ is an isolated node in $\mathcal{G}_{\mathcal{W}\setminus S}$, and so trivially, $S$ must separate $u$ and $v$, and we are done. In the case where $i \neq u$, we can manipulate the sets $U$ and $V$ so that $\boldsymbol{\omega}^{(i)}(-i)$ is not identically zero for any $i \in U, i \neq u$. If there is some $i' \in U$, $i' \neq u$, such that $X'_i \perp X_h \mid \boldsymbol{X}_S$ for all $h \in U$, $h \neq i'$, then we can simply move $i'$ from $U$ into $V$ to form a new partition $(U', V')$ of $\mathcal{W} \setminus S$. This new partition still satisfies $u \in U', v \in V'$, and $X_i \perp X_j \mid \boldsymbol{X}_S$ for all $i \in U'$ and $j \in V'$. We can therefore shift nodes one by one over from $U$ to $V$ until either $|U| = 1$, or for any $i \in U, i \neq u$, there exists an $h \in U$ such that $X_i \not\perp X_h \mid \boldsymbol{X}_S$. By the global Markov property, this assumption implies that every node $i \in U, i \neq u$ is connected by a path to some node in $U$, which means it must connected to some node in $\mathcal{W} \setminus (S \cup \{i\})$ by an edge. Thus, for all $i \in U, i \neq u$, the vector $\boldsymbol{\omega}^{(i)}(-i)$ is non-zero.

*Step 3:* We can express the conditional independence relations in terms of elements in the precision matrix $\boldsymbol{\Omega}$, since the topology of $\mathcal{G}$ can be read off the non-zero entries of $\boldsymbol{\Omega}$. The proof of the following Lemma 1 uses the matrix block inversion formula and we omit the proof due to space.

**Lemma 1** $X_i \perp X_j \mid \boldsymbol{X}_S$ if and only if $|\boldsymbol{\Omega}'(-i, -j)| = 0$.

From Lemma 1, observe that the conditional independence relations $X_i \perp X_j \mid \boldsymbol{X}_S$ are all statements about the cofactors of the matrix $\boldsymbol{\Omega}'$. It follows immediately from Lemma 1 that the vector

sets $\{\boldsymbol{\omega}^{(h)}(-i) : h \in \mathcal{W} \setminus S, h \neq j\}$ are linearly dependent for all $i \in U$ and $j \in V$. Each of these vector sets consists of the $i$-th entry truncated column vectors of $\mathbf{\Omega}'$, with the $j$-th column vector excluded. Assume that the matrix $\mathbf{\Omega}'$ is partitioned as follows,

$$\mathbf{\Omega}' = \begin{bmatrix} \mathbf{\Omega}_{UU} & \mathbf{\Omega}_{UV} \\ \mathbf{\Omega}_{VU} & \mathbf{\Omega}_{VV} \end{bmatrix}. \tag{2}$$

The strategy of this proof is to use these linear dependencies to show that the submatrix $\mathbf{\Omega}_{VU}$ has to be zero. This would imply that for any node in $U$, it is not connected to any node in $V$ by an edge. Therefore, $S$ is a node separator of $u$ and $v$ in $\mathcal{G}$, which is our goal.

*Step 4:* Let us fix $i \in U$. Consider the vector sets of the form $\{\boldsymbol{\omega}^{(h)}(-i) : h \in \mathcal{W} \setminus S, h \neq j\}$, $j \in V$. There are $|V|$ such sets. The intersection of these sets is the vector set $\{\boldsymbol{\omega}^{(h)}(-i) : h \in U\}$. We want to use the $|V|$ linearly dependent vector sets to say something about the linear dependency of $\{\boldsymbol{\omega}^{(h)}(-i) : h \in U\}$. With that in mind, we have the following lemmas.

**Lemma 2** *The vector set $\{\boldsymbol{\omega}^{(h)}(-i) : h \in U\}$ is linearly dependent for any $i \in U$.*

*Step 5:* Our final step is to show that these linear dependencies imply that $\mathbf{\Omega}_{UV} = 0$. We now have $|U|$ vector sets $\{\boldsymbol{\omega}^{(h)}(-i) : h \in U\}$ that are linearly dependent. These sets are truncated versions of the vector set $\{\boldsymbol{\omega}^{(h)} : h \in U\}$, and they are specifically truncated by taking out entries only in $U$ and not in $V$. The set $\{\boldsymbol{\omega}^{(h)} : h \in U\}$ must be linearly independent since $\mathbf{\Omega}'$ is invertible. Observe that the entries of $\mathbf{\Omega}_{VU}$ are contained in $\{\boldsymbol{\omega}^{(h)}(-i) : h \in U\}$ for all $i \in U$. We can now use these vector sets to say something about the entries of $\mathbf{\Omega}_{VU}$.

**Lemma 3** *The vector components $\boldsymbol{\omega}_j^{(i)} = \mathbf{\Omega}_{ij}$ are zero for all $i \in U$ and $j \in V$.*

This implies that any node in $U$ is not connected to any node in $V$ by an edge. Therefore, $S$ separates $u$ and $v$ in $\mathcal{G}$ and the relation $\boldsymbol{X}_u \perp \boldsymbol{X}_v \mid \boldsymbol{X}_S$ is faithful. $\qquad\square$

## 4 Algorithm for Testing Unfaithfulness

In this section, we will describe a novel algorithm for testing faithfulness of a conditional independence relation $X_u \perp X_v \mid \boldsymbol{X}_S$. The algorithm tests the necessary and sufficient conditions for faithfulness, namely, that we can find a partition $(U, V)$ of $\mathcal{W} \setminus S$ such that $u \in U, v \in V$, and $X_i \perp X_j \mid \boldsymbol{X}_S$ for all $i \in U$ and $j \in V$.

**Algorithm 1 (Testing Faithfulness)** *Input covariance matrix $\mathbf{\Sigma}$.*

1. *Define new graph $\bar{\mathcal{G}} = \{\bar{\mathcal{W}}, \bar{\mathcal{E}}\}$, where $\bar{\mathcal{W}} = \mathcal{W} \setminus S$ and $\bar{\mathcal{E}} = \{(i, j) : i, j \in \mathcal{W} \setminus S, X_i \not\perp X_j \mid \boldsymbol{X}_S, i \neq j\}$.*

2. *Generate set $U$ to be the set of all nodes in $\bar{\mathcal{W}}$ that are connected to $u$ by a path in $\bar{\mathcal{G}}$, including $u$. (A breadth-first search could be used.)*

3. *If $v \in U$, there exists a path from $u$ to $v$ in $\bar{\mathcal{G}}$, output $X_u \perp X_v \mid \boldsymbol{X}_S$ as unfaithful.*

4. *If $v \notin U$, let $V = \bar{W} \setminus U$. Output $X_u \perp X_v \mid \boldsymbol{X}_S$ as faithful.*

If we consider each test of whether two nodes are conditionally independent given $\boldsymbol{X}_S$ as one step, the running time of the algorithm is the that of the algorithm used to determine set $U$. If a breadth-first search is used, the running time is $O(|\mathcal{W} \setminus S|^2|)$.

**Theorem 2** *Suppose $X_u \perp X_v \mid \boldsymbol{X}_S$. If $S$ is a node separator of $u$ and $v$ in the concentration graph, then Algorithm 1 will classify $X_u \perp X_v \mid \boldsymbol{X}_S$ as faithful. Otherwise, Algorithm 1 will classify $X_u \perp X_v \mid \boldsymbol{X}_S$ as unfaithful.*

*Proof.* If Algorithm 1 determines that $X_u \perp X_v \mid \boldsymbol{X}_S$ is faithful, that means that it has found a partition $(U, V)$ of $\mathcal{W} \setminus S$ such that $u \in U$, $v \in V$, and $X_i \perp X_j \mid \boldsymbol{X}_S$ for any $i \in U$ and

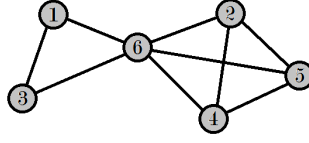

Figure 2: The concentration graph of the distribution in Example 4.

$j \in V$. By Theorem 1, this implies that $X_u \perp X_v \mid \boldsymbol{X}_S$ is faithful and so Algorithm 1 is correct. If Algorithm 1 decides that $X_u \perp X_v \mid \boldsymbol{X}_S$ is unfaithful, it does so by finding a series of nodes $w_{\ell_1}, \ldots, w_{\ell_t} \in \mathcal{W} \setminus (S \cup \{u, v\})$ for some natural number $t > 0$ such that $X_u \not\perp X_{w_{\ell_1}} \mid \boldsymbol{X}_S$, $X_{w_{\ell_1}} \not\perp X_{w_{\ell_2}} \mid \boldsymbol{X}_S, \ldots, X_{w_{\ell_{t-1}}} \not\perp X_{w_{\ell_t}} \mid \boldsymbol{X}_S, X_{w_k} \not\perp X_v \mid \boldsymbol{X}_S$, where $\ell_1, \ldots, \ell_t$ are $t$ distinct indices from $R$. By the global Markov property, this means that $u$ is connected to $v$ by a path in $\mathcal{G}$, so this implies that $X_u \perp X_v \mid \boldsymbol{X}_S$ is unfaithful and Algorithm 1 is correct. $\qquad\square$

**Example 2 (Testing an Unfaithful Distribution (1))** *Let us take a look again at the 4-dimensional Gaussian distribution in Example 1. Suppose we want to test if $X_1 \perp X_3 \mid X_2$ is faithful or not. From its covariance matrix, we have $\boldsymbol{\Sigma}_{14} - \boldsymbol{\Sigma}_{12}\boldsymbol{\Sigma}_2^{-1}\boldsymbol{\Sigma}_{24} = 2 - 2 \cdot 1/4 = 3/2 \neq 0$, so this implies that $X_1 \not\perp X_4 \mid X_2$. Similarly, $X_3 \not\perp X_4 \mid X_2$. So there exists a path from $X_1$ to $X_3$ in $\mathcal{G}_{\{1,3,4\}}$ (it is trivially the edge $(1,3)$), so the relation $X_1 \perp X_3 \mid X_2$ is unfaithful.*

**Example 3 (Testing an Unfaithful Distribution (2))** *Consider a 6-dimensional Gaussian distribution $\boldsymbol{X} = (X_1, \ldots, X_6)$ that has the covariance matrix*

$$\boldsymbol{\Sigma} = \begin{bmatrix} 7 & 1 & 2 & 2 & 3 & 4 \\ 1 & 8 & 2 & 1 & 2.25 & 3 \\ 2 & 2 & 10 & 4 & 3 & 8 \\ 2 & 1 & 4 & 9 & 1 & 6 \\ 3 & 2.25 & 3 & 1 & 11 & 9 \\ 4 & 3 & 8 & 6 & 9 & 12 \end{bmatrix}. \tag{3}$$

*We want to test if the relation $X_1 \perp X_2 \mid X_6$ is faithful or unfaithful. Working out the necessary conditional independence relations to obtain $\bar{\mathcal{G}}$ with $S = \{6\}$, we observed that $(1,3), (3,5), (5,4), (4,2) \in \bar{\mathcal{E}}$ This means that $2$ is reachable from $1$ in $\bar{\mathcal{G}}$, so the relation is unfaithful. In fact, the concentration graph is the complete graph $K_6$, and $6$ is not a node separator of $1$ and $2$.*

**Example 4 (Testing a Faithful Distribution)** *We consider a 6-dimensional Gaussian distribution $\boldsymbol{X} = (X_1, \ldots, X_6)$ that has a covariance matrix which is similar to the distribution in Example 3,*

$$\boldsymbol{\Sigma} = \begin{bmatrix} 7 & 1 & 2 & 2 & 3 & 4 \\ 1 & 8 & 2 & 1 & 2.25 & 3 \\ 2 & 2 & 10 & 4 & 6 & 8 \\ 2 & 1 & 4 & 9 & 1 & 6 \\ 3 & 2.25 & 6 & 1 & 11 & 9 \\ 4 & 3 & 8 & 6 & 9 & 12 \end{bmatrix}. \tag{4}$$

*Observe that only $\boldsymbol{\Sigma}_{35}$ is changed. We again test the relation $X_1 \perp X_2 \mid X_6$. Running the algorithm produces a viable partition with $U = \{1, 3\}$ and $V = \{2, 4, 5\}$. This agrees with the concentration graph, as shown in Figure 2.*

We include now an algorithm that learns the topology of a class of (possibly) unfaithful Gaussian graphical models using local patches. Let us fix a natural number $K < n - 2$. We consider graphical models that satisfy the following assumption: for any nodes $i$ and $j$ that are not connected by an edge in $\mathcal{G}$, there exists a vertex set $S$ with $|S| \leq K$ such that $S$ is a vertex separator of $i$ and $j$. Certain graphs have this property, including graphs with bounded degree and some random graphs with high probability, like the Erdös-Renyi graph. The following algorithm learns the edges of a graphical model that satisfies the above assumptions.

**Algorithm 2 (Edge Learning)** *Input covariance matrix $\boldsymbol{\Sigma}$. For each node pair $(i, j)$,*

1. *Let $F = \{S \subset \mathcal{W} \setminus \{i, j\} : |S| = K, X_i \perp X_j \mid \boldsymbol{X}_S, \text{ and it is faithful}\}$.*

2. *If $F \neq \phi$, output $(i, j) \notin \mathcal{E}$. If $F = \phi$, output $(i, j) \in \mathcal{E}$.*

3. *Output $\mathcal{E}$.*

Again, considering a computation of a conditional independence relation as one step, the running time of the algorithm is $O(n^{K+4})$. This comes from exhaustively checking through all $\binom{n-2}{K}$ possible separation sets $S$ for each of the $\binom{n}{2}$ $(i, j)$ pairs. Each time there is a conditional independence relation, we have to check for faithfulness using Algorithm 1, and the running time for that is $O(n^2)$. The novelty of the algorithm is in its ability to learn graphical models that are unfaithful.

**Theorem 3** *Algorithm 2 recovers the concentration graph $\mathcal{G}$.*

*Proof.* If $F \neq \phi$, $F$ is non-empty so there exists an $S$ such that $X_i \perp X_j \mid \boldsymbol{X}_S$ is faithful. Therefore, $S$ separates $i$ and $j$ in $\mathcal{G}$ and $(i, j) \notin \mathcal{E}$. If $F = \phi$, then for any $S \subseteq \mathcal{W}, |S| \leq K$, we have either $X_i \not\perp X_j \mid \boldsymbol{X}_S$ or $X_i \perp X_j \mid \boldsymbol{X}_S$ but it is unfaithful. In both cases, $S$ does not separate $i$ and $j$ in $\mathcal{G}$, for any $S \subseteq \mathcal{W}, |S| \leq K$. By the assumption on the graphical model, $(i, j)$ must be in $\mathcal{E}$. This shows that Algorithm 2 will correctly output the edges of $\mathcal{G}$. $\qquad\square$

## 5 Conclusion

We have presented an equivalence condition for faithfulness in Gaussian graphical models and an algorithm to test whether a conditional independence relation is faithful or not. Gaussian distributions are special because its conditional independence relations depend on its covariance matrix, whose inverse, the precision matrix, provides us with a graph structure. The question of faithfulness in other Markov random fields, like Ising models, is an area of study that has much to be explored. The same questions can be asked, such as when unfaithful conditional independence relations occur, and whether they can be identified. In the future, we plan to extend some of these results to other Markov random fields. Determining statistical guarantees is another important direction to explore.

## 6 Appendix

### 6.1 Proof of Lemma 2

Case 1: $|V| = 1$. In this case, $|U| > 1$ since $|U|$ and $|V|$ cannot both be one. the vector set $\{\boldsymbol{\omega}^{(h)}(-i) : h \in \mathcal{W} \setminus S, h \neq j\}$ is the vector set $\{\boldsymbol{\omega}^{(h)}(-i) : h \in U\}$.

Case 2: $|V| > 1$. Let us fix $i \in U$. Note that $\boldsymbol{\omega}^{(i)}(-j) \neq \boldsymbol{0}$ for all $j \in \mathcal{W} \setminus (S \cup \{i\})$, since the diagonal entries of a positive definite matrix are non-zero, that is, $\omega_i^{(i)} \neq 0$. Also, $\boldsymbol{\omega}^{(i)}(-i) \neq \boldsymbol{0}$ for all $i \in U$ as well by Step 2 of the proof of Theorem 1. As such, the linear dependency of $\{\boldsymbol{\omega}^{(h)}(-i) : h \in \mathcal{W} \setminus S, h \neq j\}$ for any $i \in U$ and $j \in V$ implies that there exists scalars $c_1^{(i,j)}, \ldots,$ $c_{j-1}^{(i,j)}, c_{j+1}^{(i,j)}, \ldots, c_{|\mathcal{W} \setminus S|}^{(i,j)}$ such that

$$\sum_{1 \leq h \leq |\mathcal{W} \setminus S|, h \neq j} c_h^{(i,j)} \boldsymbol{\omega}^{(h)}(-i) = 0. \tag{5}$$

If $c_i^{(i,j)} = 0$, the vector set $\{\boldsymbol{\omega}^{(h)}(-i) : 1 \leq h \leq |\mathcal{W} \setminus S|, h \neq u, j\}$ is linearly dependent. This implies that the principal submatrix $\boldsymbol{\Omega}'(-i, -i)$ has zero determinant, which contradicts $\boldsymbol{\Omega}'$ being positive definite. Thus, we have $c_i^{(i,j)} \neq 0$ for all $i \in U$ and $j \in V$. For each $i \in U$ and $j \in V$, this allows us to manipulate (5) such that $\boldsymbol{w}^{(i)}(-i)$ is expressed in terms of the other vectors in (5).

More precisely, let $\bar{\boldsymbol{c}}^{(i,j)} = [c_i^{(i,j)}]^{-1}(c_1^{(i,j)}, \ldots, c_{i-1}^{(i,j)}, c_{i+1}^{(i,j)}, \ldots, c_{j-1}^{(i,j)}, c_{j+1}^{(i,j)}, \ldots, c_{|\mathcal{W} \setminus S|}^{(i,j)})$, for $i \in U$ and $j \in V$. Note that $\boldsymbol{\Omega}'(-j, -\{i, j\})$ has the form $[\boldsymbol{\omega}^{(1)}(-i), \ldots, \boldsymbol{\omega}^{(i-1)}(-i), \boldsymbol{\omega}^{(i+1)}(-i), \ldots,$ $\boldsymbol{\omega}^{(j-1)}(-i), \boldsymbol{\omega}^{(j+1)}(-i), \ldots, \boldsymbol{\omega}^{(|\mathcal{W} \setminus S|)}(-i)]$, where the vectors in the notation described above are column vectors. From (5), for any distinct $j_1, j_2 \in V$, we can generate equations

$$\boldsymbol{\omega}^{(i)}(-i) = \boldsymbol{\Omega}'(-j_1, -\{i.j_1\})\bar{\boldsymbol{c}}^{(i,j_1)} = \boldsymbol{\Omega}'(-j_2, -\{i, j_2\})\bar{\boldsymbol{c}}^{(i,j_2)}, \tag{6}$$

or effectively,

$$\mathbf{\Omega}'(-j_1, -\{i.j_1\})\bar{\boldsymbol{c}}^{(i,j_1)} - \mathbf{\Omega}'(-j_2, -\{i, j_2\})\bar{\boldsymbol{c}}^{(i,j_2)} = 0. \tag{7}$$

This is a linear equation in terms of the column vectors $\{\boldsymbol{\omega}^{(h)}(-i) : h \neq i, h \in \mathcal{W}\}$. These vectors must be linear independent, otherwise $|\mathbf{\Omega}'(-i, -i)| = 0$. Therefore, the coefficient of each of the vectors must be zero. Specifically, the coefficient of $\boldsymbol{\omega}^{(j_2)}(-i)$ in 7 is $c_{j_2}^{(i,j_1)}/c_i^{(i,j_1)}$ is zero, which implies that $c_{j_2}^{(i,j_1)}$ is zero, as required. Similarly, $c_{j_1}^{(i,j_2)}$ is zero as well. Since this holds for any $j_1, j_2 \in V$, this implies that for any $j \in V$, $c_h^{(i,j)} = 0$ for all $h \in V, h \neq j$.

There are now two cases to consider. The first is where $|U| = 1$. Here, $i = u$. Then, by (5), $c_h^{(u,j)} = 0$ for all distinct $j, h \in V$ implies that $\omega^u(-u) = 0$, which is a contradiction. Therefore $|U| \neq 1$, so $|U|$ must be greater than 1. We then substitute $c_h^{(i,j)} = 0$, for all distinct $j, h \in V$, into (5) to deduce that $\{\boldsymbol{\omega}^{(h)}(-i) : h \in U\}$ is indeed linearly dependent for any $i \in U$.

$\square$

## 6.2 Proof of Lemma 3

Let $|U| = k > 1$ We arrange the indices of the column vectors of $\mathbf{\Omega}'$ so that $U = \{1, \ldots, k\}$. For each $i \in U$, since $\{\boldsymbol{\omega}^{(h)}(-i) : h \in U\}$ is linearly dependent and $\{\boldsymbol{\omega}^{(h)} : h \in U\}$ is linearly independent, there exists a non-zero vector $\boldsymbol{d}^{(i)} = (d_1^{(i)}, \ldots, d_k^{(i)}) \in \mathbb{R}^k$ such that $\sum_{h=1}^k d_i^{(i)} \boldsymbol{\omega}^{(h)}(-i) = 0$.

Let $\boldsymbol{y}^{(i)} = (\omega_i^{(1)}, \ldots, \omega_i^{(k)}) \in \mathbb{R}^k$. Note that $\boldsymbol{y}^{(i)} = \omega_U^{(i)}$, since $\mathbf{\Omega}'$ is symmetric, and so is a non-zero vector for all $i = 1, \ldots, k$. Because $\boldsymbol{\omega}^{(1)}, \ldots, \boldsymbol{\omega}^{(k)}$ are linearly independent, for each $i = 1, \ldots, k$, we have $\boldsymbol{d}^{(i)} \cdot \boldsymbol{y}^{(h)} = 0$ for all $h \neq i, h \in U$ and $\boldsymbol{d}^{(i)} \cdot \boldsymbol{y}^{(i)} \neq 0$.

We next show that vectors $\boldsymbol{d}^{(1)}, \ldots, \boldsymbol{d}^{(k)}$ are linearly independent. Suppose that they are not. Then there exists some index $i \in U$ and scalars $a_1, \ldots, a_{i-1}, a_{i+1}, \ldots, a_k$ not all zeros, such that $\boldsymbol{d}^{(i)} = \sum_{1 \leq j \leq k, j \neq i} a_j \boldsymbol{d}^{(j)}$. We then have $0 \neq \boldsymbol{d}^{(i)} \cdot \boldsymbol{y}^{(i)} = \sum_{1 \leq h \leq k, j \neq i} a_h \boldsymbol{d}^{(j)} \cdot \boldsymbol{y}^{(i)} = 0$, a contradiction. Therefore, $\boldsymbol{d}^{(1)}, \ldots, \boldsymbol{d}^{(k)}$ are linearly independent.

For each $j$ such that $k+1 \leq j \leq |\mathcal{W} \backslash S|$ (that is, $j \in V$), let us define $\boldsymbol{y}_j = (\omega_j^{(1)}, \ldots, \omega_j^{(k)})$. Let us fix $j$. Observe that $\boldsymbol{d}^{(h)} \cdot \boldsymbol{y}_j = 0$ for all $h = 1, \ldots, k$. Since $\boldsymbol{d}^{(1)}, \ldots, \boldsymbol{d}^{(k)}$ are linearly independent, this implies that $\boldsymbol{y}_j$ is the zero vector. Since this holds for all $j$ such that $k + 1 \leq j \leq |\mathcal{W} \backslash S|$, therefore, $\omega_j^{(i)} = 0$ for all $1 \leq i \leq k$ and $k + 1 \leq j \leq |\mathcal{W} \backslash S|$. $\square$

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
