[Reviews · NeurIPS 2014]

Submitted by Assigned_Reviewer_15

Identifying conditional independence of variables in graphical models is a key to finding tractable solutions, and faithfulness is a key condition of these relationships. The authors provide necessary and sufficient conditions for determining faithfulness in Gaussian graphical models (based on partitioning variables outside the conditioning set into two disjoint subsets), and show how this theoretical result can be translated into an algorithm for determining if a distribution is faithful.

PROS:
- Clear, well-written paper with illustrative examples
- Addresses a relevant problem and provides a meaningful theoretical result
- Provides a practical test for faithfulness in Gaussian graphical model

CONS:
- Theoretical result is restricted to Gaussian graphical models

Quality: This paper provides addresses a theoretical problem (faithfulness in Gaussian graphical models). The claims are well-reasoned and the proofs support the claims. The resulting algorithm provides a useful test of faithfulness. Although the claims are extremely specific, the results are well-reasoned.

Clarity: This paper is very well written (although minor grammatical issues), logically organized, and effectively introduces major concepts, illustrates ideas with examples. The proofs are clear and easy to follow.

Originality: I am unfamiliar of any work that addresses this particular problem, but I do not believe the analysis is particularly innovative either.

Significance: The specificity of this theoretical result (Gaussian graphical models) limits the significance of the paper, but this paper will be very useful to the particular community it targets.
Summary: This paper is well-written, makes clear claims and supports them with digestible proofs, and provides a practical test for faithfulness Gaussian graphical models. Although the audience for this paper may be limited, this paper provides a useful result for the community.

Submitted by Assigned_Reviewer_24

In this paper,the authors provided an algorithm to test the faithfulness of a conditional independence relationship in Gaussian graphical models (a faithful independence is one that is reflected in the Markov graph), which is the first of this kind. This algorithms depends on local patches of covariance matrix and requires no restrictions on the Gaussian models.

The paper is well written and clearly structured. Notation is defined early on and illustrative examples are given for important or difficult concepts(in particular Example 1 and Example 4). Also, intuitions are given for conclusions that are difficult to parse by themselves (e.g. Theorem 1). The theorems and algorithms are novel and the proofs are correct. I enjoyed reading this paper and recommend publication.

I have a few suggestions on the final version:

1) The authors should provide additional complexity analysis for Algorithm 1.

2) While Theorem 2 is correct, it can be restated for clarity. Basically, it's saying that Algorithm is correct.

Summary: In this paper, the authors have provided a novel algorithm to test the faithfulness of a Gaussian graphical model. I recommend publication.

Submitted by Assigned_Reviewer_31

This paper considers testing for faithfulness in (undirected) Gaussian Graphical Models, i.e. the condition where X ind Y | Z in the probability distribution => X separates Y | Z in the GGM. The authors prove a result which characterizes unfaithful conditional independence relations in GGMs and leads to an algorithm for testing whether a conditional independence relation, and hence distribution, is unfaithful, which the authors prove is correct and provide a few examples for.

The paper is well written and clear. It is original to my knowledge; I'm not aware of any other papers for testings faithfulness in GGMs. The technical work appears to be sound (though I did not check all proofs thoroughly). The motivation/significance is somewhat lacking, however. The result is theoretically interesting, but it's not clear to me when faithfulness checking in GGMs would be of practical use. I can see how this would be useful if one was using constraint-based methods to learn GGMs, but I don't see how this would improve state of the art methods based on maximizing the multivariate normal likelihood with an l1 penalized precision matrix. Since the authors haven't provided any other significant motivation, it's not clear to me how this result may be useful. Furthermore, the algorithm the authors provide requires many conditional independence tests just to test the faithfulness of one conditional independence relation and hence is unlikely to be robust (no empirical results are provided).
Summary: This paper characterizes unfaithful conditional independence relations in (undirected) Gaussian Graphical Models and provides an algorithm for testing whether a conditional independence relation is faithful. The paper is well written, sound, and original, but does not provide much motivation for the result or give examples of instances where it would be useful (in learning GGMs or elsewhere).
Author Feedback
Author rebuttal: We thank the reviewers for their reviews and comments. We appreciate that the reviewers acknowledged the novelty of our work. We hope to be able to address adequately some of the concerns raised by the reviewers. These concerns can be divided into two main categories.

1) "Theoretical result is restricted to Gaussian graphical models":

Gaussian graphical models are widely used in many applications. Also, Gaussian graphical models have the property that the graphs used to present these models are determined by the non-zero entries of their inverse covariance matrices. Other graphical models may not have such a property. An Ising model, for example, has a graph that is not reflected entirely by its inverse covariance matrix. Loh and Wainwright explored this in their paper "Structure estimation for discrete graphical models: Generalized covariance matrices and their inverses"; the inverse covariance matrix of an Ising model reflects the graph structure only for certain graph topologies, like trees. Testing the faithfulness of more general graphical models thus requires new techniques and is an open problem.

2) "The paper... does not provide much motivation for the result or give examples of instances where it would be useful (in learning GGMs or elsewhere)":

In learning Gaussian graphical models, many authors, as cited in our paper, make use of conditional independence relations to learn the topology of the graph. This fundamentally relies on the graphical model being faithful, because unfaithfulness introduces ambiguity into the topological interpretation of conditional independence relations. As such, the work that has been done in this area of using conditional independences have restrictions on their models that overcome the issue of unfaithfulness. Some of these restrictions include walk-summability, sparsity, minimum girth, and the number of feedback vertices. A consequence of placing some of these restrictions is it overcomes the problem of unfaithfulness. Therefore, our method allows for these restrictions to be removed by testing the faithfulness of conditional independence relations. This allows us to learn more general Gaussian models. We would like to highlight that the only assumption we place on the graphical model is that it must be Gaussian. If it is Gaussian, we can test the faithfulness of its conditional independence relations.
An example of a paper that places restrictions on Gaussian graphical models is the paper "High-Dimensional Gaussian Graphical Model Selection: Walk-Summability and Local Separation Criterion" by Anandkumar et al.
We note the reviewers comment on comparisons with maximizing the multivariate normal likelihood with l1 constraints. These methods usually require some form of sparsity to be presented. Introducing sparsity intuitively reduces the chance of a model being unfaithful. Our result is a step towards learning Gaussian graphical models that have no such restriction. However, there is still great value in comparing conditional independence techniques against regularization techniques in learning graphical models, which is beyond the scope of our work in this paper. We are currently exploring this in future work.

Finally, we list some of the future changes we will be making to this paper. We will include the computational complexity for this algorithm. We will also restate Theorem 2 in a clearer fashion.